# Effects of LED light spectra and intensity on winter citrus nursery production

**Rayane Barcelos Bisi[1,2], Ute Albrecht[1], Randall P. Niedz [2]\*, Kim D. Bowman [2]**

**1** Southwest Florida Research and Education Center, University of Florida/IFAS, Immokalee, Florida, United States of America, **2** U.S. Horticultural Research Laboratory, USDA, ARS, Fort Pierce, Florida, United States of America

\* randall.niedz@usda.gov

## Abstract

This study investigated the effects of supplemental light-emitting diode (LED) light spectra and intensity on the winter growth of grafted citrus nursery plants. The primary objective was to determine how different combinations of red, blue, white, and far-red LED light, applied at varying intensities, influence vegetative budbreak and scion shoot growth in young citrus trees during short winter days. The experiment used two citrus rootstocks, Carrizo (a hybrid of sweet orange and trifoliate orange) and Rich 16−6 (trifoliate orange), each budded with Washington Navel orange. Plants were grown in a temperature-controlled greenhouse and received natural sunlight, with supplemental LED lighting used to extend the photoperiod to 16 hours. The study employed a 4-factor response surface design, testing multiple combinations of light spectra and intensity. Eight plant growth and physiological responses were measured, including percentage budbreak, scion shoot length and diameter, rootstock diameter, internode length, leaf area, scion dry weight, and leaf chlorophyll index. Results showed that both the quality and intensity of supplemental LED light significantly affected all measured responses. Blue and white light, particularly at higher intensities, increased vegetative budbreak in both rootstocks, while red and far-red light reduced budbreak. The greatest scion shoot growth was observed when red, blue, and white light were combined at maximum intensity. Far-red light alone did not enhance growth, but when combined with red or white light, it further increased shoot elongation and leaf area. Chlorophyll content was highest under blue and white light and lowest under far-red light. These findings demonstrate that optimizing supplemental LED light spectra and intensity can improve winter nursery production of grafted citrus by enhancing budbreak and shoot growth. The results provide a basis for refining light management strategies in commercial citrus nurseries to increase propagation efficiency during periods of limited natural light.

**Data availability statement:** All relevant data are within the manuscript and its Supporting Information files.

**Funding:** This research was conducted at the United States Horticultural Research Laboratory, Ft Pierce, Florida, USA and was funded by the California Citrus Nursery Board (ALB-21-23 and BOW-21-23). The funders had no role in study design, data collection and analysis, decision to publish, or preparation of the manuscript.

**Competing interests:** The authors have declared that no competing interests exist.

## Introduction

Citrus production in commercial nurseries relies on the propagation (grafting) of scion cultivars on selected rootstocks, a process that is sensitive to environmental conditions during the winter months. Reduced daylength and lower temperatures during this period limits vegetative budbreak and the subsequent growth of grafted scions, thereby extending the nursery production cycle and potentially affecting overall productivity. To address these limitations, supplemental lighting was employed to extend the daylength and improve plant growth during the winter, with several studies in citrus and other horticultural crops demonstrating positive effects on budbreak and shoot development [1–8].

Light is a fundamental environmental factor that regulates plant physiology throughout the life cycle, functioning as both an energy source for photosynthesis and a signal that controls a range of developmental processes [9]. The response of plants to light is complex and depends on several parameters, including intensity, photoperiod, and spectral composition [10–12]. Recent advances in LED technology have made it possible to manipulate light spectra with precision, enabling the study of specific wavelength effects-such as blue, red, and far-red light-on plant growth and development [13]. While photosynthetically active radiation (PAR) is traditionally defined as 400–700 nm, far-red light (700–800 nm) has a role in regulating phytochrome-mediated responses that influence plant morphology and development [10,14–16].

In controlled environments, supplemental lighting is commonly used to achieve a target daily light integral (DLI), particularly in regions or seasons with limited natural sunlight. However, both insufficient and excessive light intensity can have adverse effects: low intensity may not support optimal growth, while high intensity can induce stress responses, including the production of antioxidants and non-photosynthetic pigments to mitigate photodamage [17–19]. Plants have evolved diverse photoprotection mechanisms to optimize photosynthesis and prevent photoinhibition under fluctuating light conditions.

The literature indicates that plant responses to supplemental light vary widely among species and even among cultivars within a species [20–22]. While the effects of light spectra and intensity have been extensively studied in various horticultural crops, there is little information in the citrus literature regarding the interactive effects of these factors on budbreak and early growth. This knowledge is essential for optimizing nursery production practices, particularly during the winter when natural light is limited.

The role of far-red light in plant growth is not completely understood. Although far-red photons are inefficient at driving photosynthesis when applied alone, their inclusion with other wavelengths can enhance overall photochemical efficiency – a phenomenon known as the Emerson enhancement effect [15–16,23]. The practical implication for citrus nursery production of the interaction between far-red light with the other wavelengths was considered when planning this study

Given the complexity of plant responses to light, a multivariate approach is necessary to disentangle the effects of spectral quality and intensity as well as identify

optimal supplemental lighting strategies. In citrus nurseries, where winter light limitations delay graft maturation, LED systems can provide the specific spectra to enhance budbreak as observed in preliminary trials [3]. In this study, we quantify how specific combinations and intensities of red, blue, white, and far-red LED light affect budbreak and early scion growth of grafted citrus during winter, when natural light is limited. Our objective was to provide nursery managers with information for using supplemental lighting to shorten production time, improve plant quality, and enhance propagation efficiency in commercial citrus nurseries.

## Materials and methods

### Plant material and growth conditions

Citrus rootstock seedlings were cultivated using two varieties that exhibit a high frequency of nucellar polyembryony and apomictic seedlings: Carrizo (*Citrus sinensis × Poncirus trifoliata*) and Rich 16−6 (*P. trifoliata*). Individual seeds were planted in cone-shaped containers measuring 3.8 cm in diameter and 21 cm in height (Cone-tainers; Stuewe and Sons, Tangent, OR, USA). These containers were filled with a soilless growing medium (Pro Mix BX; Premier Horticulture, Inc., Quakertown, PA, USA). Seedlings exhibiting atypical characteristics were identified through visual inspection of their morphology and subsequently removed from the study. Because commercial citrus rootstock production relies on nucellar polyembryony as the source of rootstock seed, citrus nurseries routinely remove seedlings exhibiting clear morphological deviations –typically, those less than 30% the height of typical seedlings or displaying aberrant leaf characteristics [24]. The remaining true-to-type seedlings were then transplanted into larger 2.54 L pots (Treepots; Stuewe and Sons) using the same soilless growing medium. Prior to budding, the diameter of each rootstock trunk was measured 15 cm above the soil surface. This measurement was taken approximately two weeks before budding, and any plants found to be undersized were excluded from the experiment. To maintain uniformity within treatment groups, the remaining plants were categorized based on their trunk diameter and randomly assigned to different treatments.

On November 21, 2025 six months old seedlings from Carrizo and Rich 16−6 were budded with Washington Navel orange (*C. sinensis*) using the inverted T bud method. Grafted buds were wrapped with bud tape for 3 weeks. One week after unwrapping, the rootstock was trimmed at 65 cm and looped to force bud growth as described [25]. Plants were arranged on the bench with the grafted bud facing south to increase vegetative budbreak and scion shoot growth [26]. The inverted T-budding method and south-facing bud orientation were implemented with a high degree of standardization and were supported by multiple procedural and design controls. The uniform budding date, pre-budding plant selection, randomization, continuous environmental monitoring, and high replication collectively minimized the influence of micro-environmental variation on budbreak initiation.

Plants were grown in a temperature-controlled greenhouse set at 21–27 °C, controlled by heaters, cooling fans, wet pad and circulating fans. Temperature controls were set to run heating when temperature dropped below 21 °C and run cooling when temperature rose above 27 °C. Plants were irrigated to run-off every other week with water-soluble fertilizer, and as needed with water between fertilization treatments. The fertilizer used was 20N-10P-20K (Peters Professional, The Scotts Company, Marysville, OH) at a rate of 400 mg N/L. Insecticides and miticides were applied as needed.

### LED light treatments

Plants received natural sunlight during the day, with supplemental LED lighting provided in the early morning hours to achieve a total photoperiod of 16 hours. The duration of supplemental lighting was calculated daily as the difference between the 16-hour target photoperiod and the natural daylength, which increased gradually over the 12-week experimental period. For example, when natural daylength was 10 hours, the LED fixtures operated for 6 hours before sunrise; as natural daylength increased to 11 hours, LED operation was reduced to 5 hours. This approach maintained a consistent 16-hour photoperiod throughout the experiment while allowing plants to receive the full duration of available natural sunlight each day.

 

The light sources used were light-emitting diode (LED) Elixia 600W fixtures (Heliospectra, Göteborg, Sweden). Each fixture contained four LED types: 1) Red – peak at 660 nm; 2) Blue – peak at 450 nm; 3) White 5700K – peak at 446 nm, 534 nm, and 625 nm; and 4) Far-red – peak at 735 nm. The intensity of each LED type was varied from 0 to 1,000, the maximum intensity for each wavelength of the fixture. Heliospectra LED lights use a relative, unitless scale for intensity control in their software.

Light fixtures were positioned 110 cm above the bud insertion point to ensure uniform photosynthetic photon flux density (PPFD) distribution among plants under each light treatment. LED timing was adjusted throughout the 12-week experimental period in response to changing natural daylength: whenever natural daylength changed by 15 minutes, the supplemental lighting schedule was recalculated and adjusted to maintain the 16-hour total photoperiod. Light energy and the average PPFD under each LED light were measured using a Spectrometer PG100N (UPRtek, Zhunan, Taiwan) for each treatment at 110 cm below the light fixture, at bud insertion level, and on a moonless night.

### Air temperature

A HOBO data logger, model MX2203 (Onset Computer Corporation, MA, USA), was installed among the plants in each treatment and the air temperature was recorded every 10 minutes from the date of budding to 12 weeks-after-budding (wab).

### Leaf surface temperature

Measurements of the leaf surface temperature were taken at 12 wab with a hand-held infrared radiometer MI-210 (Apogee Instruments, Logan, UT, USA). The leaf temperature was measured at night, on the first fully expanded leaf from the scion growth on seven different plants under the light in each treatment.

### Daily light integral (DLI)

Photosynthetic active radiation (PAR) is the wavelengths of light within the visible range of 400–700 nanometers (nm) which drive photosynthesis. The PPFD is the amount of PAR that is hitting the plant in a specific location and time. PPFD was monitored for each treatment using analog full-spectrum quantum sensors SQ-500 (Apogee Instruments, Logan, UT, USA). Each sensor was attached to a data logger and PPFD was measured every 150 seconds and averaged every 900 seconds. DLI was calculated for each treatment using the formula:

$$DLI = \frac{\Sigma PPFD \times 900}{1,000,000}$$

### Responses measured

Eight responses were measured as follows:

1) Budbreak % – Percentage of vegetative buds that had formed shoots by 12 wab (scion bud initiation: shoot tip of bud at least 2 mm long).

2) Bud shoot growth – Length (mm) of bud shoots at 12 wab.

3) Scion diameter – Diameter (mm) of the bud shoot 5 cm above the graft union 12 wab.

4) Rootstock diameter – Rootstock stem diameters (mm) were measured at 5 cm below the graft union at 12 wab.

5) Internode length – Scion length was measured up to the last fully expanded leaf and divided by the number of nodes for average of internode length at 12 wab. Measured as mm.

 

6) Leaf area – Total leaf area of the bud shoot leaves (cm$^2$) at 12 wab. Only fully expanded leaves were used for leaf area measurement. The total leaf area was measured using a LI-3100C leaf area meter (LI-COR Biosciences, Lincoln, NE, USA).

7) Scion dry weight – The bud shoot was excised at 12 wab and dried in an oven at 70°C to a constant weight (g).

8) Chlorophyll index – Measured the first fully expanded leaf with a SPAD-502 portable chlorophyll meter (Minolta, Spectrum Technologies, Inc., Plainfield, IL, USA). Seven leaves from the 'Washington' Navel scion from each rootstock and light treatment were measured and averaged at 12 wab.

## Experimental design and analyses

Response surface methodology (RSM) was used to model the interactions between four LED light spectra (red, blue, white, far-red) and their intensities while also identifying optimal combinations for various horticultural growth measures. RSM enables for the simultaneous analysis of linear, quadratic, and interaction effects – necessary for understanding and optimizing responses like budbreak and shoot elongation. Unlike single-factor experiments or basic factorial designs, RSM provides greater statistical power to detect nonlinear patterns and synergistic effects, and in a more efficient approach.

The effects of supplemental LED lighting on budbreak and plant growth were evaluated in two citrus rootstocks, Carrizo and Rich 16−6, using a four-factor D-optimal response surface design (Tables 1 and 2). Experimental factors were the "intensity units" of four types of LED lights – blue, red, white, and far-red – with each power input varying from 0 to 1,000 "intensity units". The spectral power distribution (SPD) profile was generated for each design run (S1–S3 Figs). The design allowed for estimation of main effects, two-factor interactions, and quadratic effects for each of the four LED light types. The core experimental design consisted of 29 runs, including 15 model points, 5 lack-of-fit points, and 5 replicate points for pure error estimation. Four additional points of interest (Runs #3, #7, #18, and #27) were added and treated as lack-of-fit points in the analyses. In planning the study, we conducted a prospective power analysis of the selected 4-factor D-optimal RSM design assuming a residual standard deviation of 1 on the coded response scale. Under this assumption, the design had very high power (≥ 99.6% at α = 0.05) to detect the main, interaction, and quadratic effect sizes: for example, the computed powers for A, B, C, D and their two-factor interactions (AB, AC, AD, BC, BD, CD) were all 99.9%, and the powers for the quadratic terms A², B², C², and D² were 99.9%, 99.9%, 99.9%, and 99.6%, respectively. These effect sizes correspond to changes that are large relative to the within-treatment variability and are biologically meaningful for nursery performance, indicating that the design was well powered for effects of practical interest.

A summary of the entire experimental design is provided (Table 3).

For each lamp, 96 plants (48 Carrizo and 48 Rich 16−6) were grown. The area under each lamp was divided into four quadrants, with each quadrant containing 24 plants (12 of each rootstock). This arrangement resulted in each lamp illuminating 48 Washington Navel orange plants budded on Carrizo and 48 budded on Rich 16−6 rootstock. In total, 2,784 plants were used in the experiment. For statistical analysis, the response for each treatment design point and rootstock was calculated by averaging the responses of the 12 plants in each quadrant.

**Table 1. The four LED light factors and their ranges to construct the experimental design.**

| Factors | LED (peak wavelength, nm) | Range (intensity units) |
|---|---|---|
| A | Red (660) | 0–1,000 |
| B | Blue (450) | 0–1,000 |
| C | White (446, 534, 625) | 0–1,000 |
| D | Far-red (735) | 0–1,000 |

**Table 2. Four-factor D-optimal quadratic response surface design matrix. PPFD = Photosynthetic Photon Flux Density – a measure of how many photons land in a specific area that the plant can use.**

| Run | Intensity Units | | | | PPFD |
| --- | --- | --- | --- | --- | --- |
| | Red | Blue | White | Far-red | $\mu$mol m$^{-2}$ s$^{-1}$ |
| 1 | 500 | 750 | 750 | 250 | 426 |
| 2 | 0 | 500 | 0 | 1000 | 51 |
| 3 | 1000 | 0 | 0 | 0 | 352 |
| 4 | 1000 | 1000 | 500 | 1000 | 560 |
| 5 | 1000 | 0 | 0 | 1000 | 360 |
| 6 | 1000 | 1000 | 0 | 500 | 426 |
| 7 | 0 | 0 | 0 | 0 | 0 |
| 8 | 1000 | 500 | 1000 | 1000 | 628 |
| 9 | 0 | 1000 | 1000 | 1000 | 320 |
| 10 | 1000 | 1000 | 500 | 1000 | 554 |
| 11 | 0 | 1000 | 1000 | 1000 | 320 |
| 12 | 500 | 1000 | 0 | 1000 | 263 |
| 13 | 1000 | 1000 | 1000 | 0 | 664 |
| 14 | 0 | 500 | 1000 | 0 | 282 |
| 15 | 0 | 500 | 1000 | 0 | 282 |
| 16 | 1000 | 0 | 0 | 1000 | 360 |
| 17 | 500 | 250 | 750 | 750 | 392 |
| 18 | 1000 | 1000 | 1000 | 1000 | 671 |
| 19 | 1000 | 0 | 1000 | 0 | 588 |
| 20 | 0 | 0 | 0 | 0 | 0 |
| 21 | 1000 | 0 | 1000 | 0 | 588 |
| 22 | 1000 | 1000 | 0 | 0 | 430 |
| 23 | 0 | 1000 | 0 | 0 | 81 |
| 24 | 0 | 0 | 0 | 1000 | 6 |
| 25 | 0 | 0 | 1000 | 1000 | 244 |
| 26 | 500 | 250 | 250 | 250 | 275 |
| 27 | 250 | 750 | 250 | 500 | 226 |
| 28 | 750 | 500 | 250 | 750 | 387 |
| 29 | 1000 | 500 | 0 | 0 | 398 |

Analysis of variance (ANOVA) models were constructed using forward selection based on Akaike's Information Criterion (AICc) [27] to select the highest order polynomial model where additional terms were significant at the 0.05 significance level. Model diagnostics and adequacy were evaluated using multiple methods to ensure reliable statistical analyses. We evaluated whether the regression/ANOVA assumptions were reasonably satisfied before interpreting the response-surface models. Specifically, we used a Box–Cox analysis [28] to determine whether a response required a power transformation. We then checked normal probability plots of the residuals and plots of internally studentized residuals versus predicted values to assess, respectively, approximate normality and constant variance after any transformation. We also reviewed lack-of-fit tests, Cook's distance, leverage, and DFFITS plots to identify observations with undue influence on the fitted models. Finally, we evaluated overall model quality using the agreement among R², adjusted R², and predicted R², and required adequate precision values greater than 4 [29] to ensure sufficient signal-to-noise. We used these diagnostics,

**Table 3. Summary of the experimental setup for the response surface design for winter photo-period supplementation with LED lighting including A) Plant material & grafting, B) Greenhouse conditions, C) Photoperiod strategy, D) LED spectra & measurement, E) Experimental design, F) Replication & bench layout, G) Responses, and H) Statistical analysis.**

| A. Plant material & grafting | B. Greenhouse conditions (winter trial) |
|---|---|
| Rootstocks (6-mo seedlings):<br>- Carrizo<br>- Rich 16−6<br>Scion: Washington Navel orange.<br>Budding: Inverted T-bud (Nov 21, 2025)<br>- Bud tape 3 wks<br>- Trim at 65 cm & loop<br>- Bud oriented south | Temperature range: 21–27°C<br>Climate control: heater/cooling/fans<br>Irrigation + nutrition:<br>- Irrigate to runoff; water as needed<br>- 20N-10P-20K @400 mg N/L<br>Monitoring: HOBO MX2203 logging (10-min logging) |
| **C. Photoperiod strategy** | **D. LED spectra & measurement** |
| Natural sunlight during the day<br>Supplemental LEDs used pre-dawn to:<br>- Maintain 16-h photoperiod<br>- Daily LED hours = 16 h – natural daylength<br>- Schedule recalculated when daylength changed by 15 min<br>- Duration: 12 weeks after budding (12 wab) | Heliospectra Elixia 600W fixtures<br>Four controllable channels (0–1000 units):<br>- Red 660 nm<br>- Blue 450 nm<br>- White 5700K<br>- Far-red 735 nm<br>Fixture height: 110 cm above bud graft<br>PPFD/SPD measured at night<br>PPFD logging (SQ-500) -> DLI calculated |
| **E. Experimental design** | **F. Replication & bench layout** |
| Response Surface Methodology (RSM)<br>Design: 4-factor D-optimal quadratic<br>Factors: red, blue, white, far-red (0–1000 units)<br>Core: 29 runs (25 runs + 4 additional points)<br>Estimates: main effects, 2-factor interactions, and quadratic terms | Per lamp (per design run): 96 plants<br>− 48 Carrizo + 48 Rich 16−6<br>Lamp area split into 4 quadrants<br>- Each quadrant: 24 plants (12 per rootstock)<br>Analysis unit: quadrant mean per rootstock/run<br>Total plants: 2,784 |
| **G. Responses (measured at 12 wab)** | **H. Statistical analysis** |
| 1. Budbreak % (shoot ≥ 2 mm)<br>2. Scion shoot length<br>3. Scion shoot diameter<br>4. Rootstock diameter<br>5. Internode length<br>6. Leaf area<br>7. Scion dry weight<br>8. Chlorophyll index (SPAD) | Separate models per rootstock<br>Model selection: forward AICc selection<br>Diagnostics: Box-Cox, residuals, lack-of-fit<br>Influence: Cook's D; leverage, DFFITS<br>Model quality: $R^2$, adj $R^2$, pred $R^2$; adequate precision >4 |

taken together, to indicate if the models met the standard ANOVA assumptions well enough to support using for valid inferences.

The RSM framework used in this study differs from the traditional univariate approach used where each outcome is analyzed in isolation: a separate set of formal tests run for each response, and the results interpreted one outcome at a time. In contrast, the RSM approach still fits a separate model to each response but places greater emphasis on overall model adequacy—assessed via residual plots, lack-of-fit, and summary statistics such as $R^2$, adjusted $R^2$, predicted $R^2$, and adequate precision—rather than on numerous assumption tests. When multiple related outcomes are present, RSM treats them as different facets of the same experimental system and focuses on effects and patterns that are consistent across responses, rather than treating them as a collection of independent univariate analyses where multiple comparison correction is required. In this experiment the eight responses came from a single experiment and are considered all related aspects of plant performance. Because all eight responses arose from the same RSM design that varied red, blue, white, and far-red LED intensities on the same Carrizo and Rich 16−6 nursery trees, they were treated as related aspects

of plant performance. Interpretation emphasized statistically supported factor effects and trends that pointed in similar directions across responses, rather than treating each response as an independent univariate analysis.

## Results

### Temperature

The air temperature inside the greenhouse varied from 21 to 27°C, the minimum temperature average ranged from 20 to 25 °C and the maximum temperature average ranged from 23 to 34°C.

### Daily light integral (DLI)

The PPFD from daylight increased throughout the experiment and consequently, the average of DLI increased. The total average of DLI under the no supplemental light treatment was 5 mol·m$^{-2}$·day$^{-1}$. The highest DLI was under treatment 13, which had red + blue + white light at the highest intensity. The amplitude of DLI among the treatments over the twelve weeks varied from 8 to 11 mol·m$^{-2}$·day$^{-1}$.

### Vegetative and physiological measurements

In this study, the primary question was the light–response relationship within each rootstock (i.e., the response surface over the quantitative spectral factors), not simply the overall difference between Carrizo and Rich 16−6. Because rootstock is a very strong factor it would largely summarize the obvious mean shift between rootstocks while obscuring the more subtle, but biologically important, effects and interactions among the continuous light variables. The two rootstocks were analyzed separately to preserve the interpretability of the light–response surfaces and avoided over-emphasizing an inflated overall $R^2$ that would mostly represent the strong rootstock main effect rather than the patterns that are most relevant for optimizing light management.

The ANOVA results summarizing the eight plant response variables measured on the Carrizo and Rich 16−6 rootstocks are presented (Tables 4 and 5), with the corresponding datasets provided (S1 File). For each response a description is provided of the range of the data observed, the p-value of the model, the result of the Box Cox analysis and requirements for data transformation, the three R2 values as general measure of model quality, the significant terms in each model, and the model's settings that optimized a defined maximized response.

In several models, the three R² statistics were modest in absolute value, even though the adequate precision exceeded the recommended threshold of 4 and the lack-of-fit tests did not indicate that additional signal could be captured with a more complex model. This combination reflects the biological reality that much of the variability in growth traits of grafted citrus trees arises from inherent plant-to-plant and micro-environmental variation that is not reducible by adjusting light spectra alone. In practical terms, the response surface models are best interpreted as tools for identifying directional trends and optimal regions within the LED design space, rather than for making precise numerical predictions for individual plants. The fitted surfaces describe how changes in red, blue, white, and far-red intensity shift the expected mean response across treatments, but two plants grown at the same settings will still differ because of underlying biological noise. Thus, the models support decisions about which spectral combinations are likely to improve budbreak, shoot growth, and canopy development during winter nursery production, even though they explain only a fraction of the total variation observed among individual trees.

### Percentage budbreak

Bud survival on Carrizo and Rich 16−6 was 98.1% and 96.6% on average. Of the surviving buds, the percentage budbreak ranged from 8 to 92% for Carrizo and 0–58% for Rich 16−6, indicating light affected percentage budbreak. A reduced quadratic polynomial model was selected for Carrizo (p < 0.001) and Rich 16−6 (p < 0.001). Data were transformed for Rich

**Table 4. ANOVA model terms, p-values (Prob.>F), lack-of-fit, R², and adequate precision statistics for the effect of light on % budbreak, shoot growth, and scion and rootstock dimeters. Full ANOVA tables and model equations in coded and actual form provided (S2 File).**

| Source | Budbreak (%) | | Scion shoot growth (mm) | | Scion shoot diameter (mm) | | Rootstock diameter (mm) | |
|---|---|---|---|---|---|---|---|---|
| | Carrizo | Rich 16−6 | Carrizo | Rich 16−6 | Carrizo | Rich 16−6 | Carrizo | Rich 16−6 |
| Model | 0.0005 | 0.0001 | 3.41e-06 | 0.0016 | 6.029e-05 | 0.0002 | 3.49e-13 | 0.0001 |
| A – Red | 0.0025 | 0.6730 | 0.0043 | 0.1870 | 0.0089 | 0.0010 | 2.53e-05 | 8.29e-05 |
| B – Blue | 0.0155 | 0.0264 | 0.0263 | 0.4185 | 0.15355 | 0.8934 | 0.0001 | – |
| C – White | – | 0.0018 | 0.0075 | 0.0640 | 0.0316 | 0.9117 | 1.17e-07 | 0.0448 |
| D – Far-red | 0.6818 | 0.5092 | 0.5057 | 0.4990 | 0.7755 | – | 0.0145 | 0.3818 |
| AB – Red x Blue | – | – | – | – | – | 0.0996 | 0.1138 | – |
| AC – Red x White | – | – | – | – | – | 0.1527 | – | – |
| AD – Red x Far-red | 0.0160 | 0.0036 | 0.0478 | 0.0280 | 0.0369 | – | – | – |
| BC – Blue x White | – | – | 0.0012 | – | 0.0062 | – | 0.1182 | – |
| BD – Blue x Far-red | – | – | 0.1139 | – | – | – | – | – |
| CD – White x Far-red | – | – | – | – | – | – | – | 0.0603 |
| $A^2$ – $Red^2$ | 0.0613 | – | – | – | – | – | 0.0333 | – |
| $B^2$ – $Blue^2$ | – | – | – | 0.0042 | – | 0.0310 | – | – |
| $C^2$ – $White^2$ | – | – | – | – | – | – | – | – |
| $D^2$ – $Far\text{-}red^2$ | – | – | – | – | – | – | – | – |
| Lack of Fit | 0.0166 | 0.6336 | 0.0004 | 0.4902 | 0.0074 | 0.8313 | 0.1182 | 0.5650 |
| $R^2$ | 0.1733 | 0.1958 | 0.2809 | 0.1748 | 0.2248 | 0.2198 | 0.4723 | 0.1947 |
| $R^2$ adj | 0.1371 | 0.16055 | 0.2360 | 0.1293 | 0.1833 | 0.1748 | 0.4393 | 0.1643 |
| $R^2$ pred | 0.0823 | 0.1083 | 0.1871 | 0.0638 | 0.1350 | 0.1126 | 0.4019 | 0.1100 |
| Adequate precision | 7.6325 | 8.4574 | 8.1371 | 7.8005 | 8.4508 | 7.0246 | 13.8638 | 8.1618 |
| Transformation | no | square root | no | no | no | no | no | no |
| Model type | reduced quadratic | reduced 2FI | reduced 2FI | reduced quadratic | reduced 2FI | reduced quadratic | reduced quadratic | reduced 2FI |

16−6 per violation of the ANOVA assumption of normally distributed residuals as detected via Box-Cox analysis. A summary of the ANOVA, lack-of-fit test, three $R^2$ statistics, and adequate precision statistic for % budbreak is presented (Table 4). The lack-of-fit (LOF) test was significant for Carrizo (p = 0.0166) and was interpreted to be largely a consequence of small error estimates rather than an indication that additional variation could be removed with a better model, as evidenced by the pure error mean square being substantially smaller than both the lack-of-fit and model mean squares (S2 File). For both rootstocks $R^2$, $R^2_{adj}$ and $R^2_{pred}$ statistics were tightly clustered, $R^2$ - $R^2_{adj}$ and $R^2_{adj}$ - $R^2_{pred}$ < 0.2, and adequate precision was > 4 indicating adequate models for prediction. The ANOVA for Carrizo contained 3 significant terms (p < 0.05) – Red, Blue, and Red x Far-red. The ANOVA for Rich 16−6 contained three significant terms, the main effects Blue and White, and Red x Far-red. For Carrizo, the largest effects Red and Red x Far-red, which reduced predicted % budbreak, while blue

**Table 5. ANOVA model terms, p-values (Prob. > F), lack-of-fit, R², and adequate precision statistics for the effect of light on internode length, leaf area, scion biomass, and chlorophyll index. Full ANOVA tables and model equations in coded and actual form provided (S2 File).**

| Source | Internode length (mm) | | Leaf area (cm²) | | Scion dry weight (g) | | Chlorophyll index | |
|---|---|---|---|---|---|---|---|---|
| | Carrizo | Rich 16−6 | Carrizo | Rich 16−6 | Carrizo | Rich 16−6 | Carrizo | Rich 16−6 |
| Model | 0.0002 | 0.0034 | 3.09e-09 | 0.0002 | 5.23e-09 | 2.43e-07 | 2.144e-31 | 1.34e-17 |
| A – Red | 0.0462 | 0.0350 | 0.0070 | 0.0262 | 2.10e-05 | 3.55e-05 | 0.0193 | 1.49e-05 |
| B – Blue | 0.0429 | 0.9080 | 0.0026 | 0.9363 | 0.0109 | 0.9744 | 7.97e-13 | 5.10e-14 |
| C – White | 0.7427 | 0.46448 | 0.0020 | 0.0276 | 0.0028 | 0.0051 | 3.96e-15 | 0.0273 |
| D – Far-red | 0.2988 | 0.2289 | 0.9482 | 0.9280 | 0.9753 | 0.0407 | 2.05e-22 | 0.0001 |
| AB – Red x Blue | – | – | – | – | – | – | 0.0373 | – |
| AC – Red x White | – | 0.0142 | – | – | – | 0.0946 | – | – |
| AD – Red x Far-red | 0.0786 | – | 0.0018 | 0.0187 | 0.0370 | – | 2.56e-05 | – |
| BC – Blue x White | 0.0024 | 0.1719 | 0.0001 | 0.1181 | 0.0009 | – | – | – |
| BD – Blue x Far-red | – | 0.1180 | – | – | – | – | 0.0657 | 0.0236 |
| CD – White x Far-red | – | – | 0.0689 | – | – | – | 0.0002 | 0.0191 |
| A² – Red² | – | – | – | – | – | – | – | 0.0009 |
| B² – Blue² | – | 0.0418 | – | 0.0026 | – | 0.0308 | – | 0.0208 |
| C² – White² | – | – | – | – | – | – | 7.28e-10 | – |
| D² – Far-red² | – | – | – | – | – | – | – | – |
| Lack of Fit | 0.0056 | 0.2727 | 0.0002 | 0.8579 | 0.0125 | 0.2641 | 5.28e-21 | 1.14e-13 |
| R² | 0.2068 | 0.1969 | 0.3730 | 0.2369 | 0.3515 | 0.3234 | 0.7726 | 0.5758 |
| R² adj | 0.1646 | 0.1339 | 0.3339 | 0.1850 | 0.3171 | 0.2844 | 0.7539 | 0.5453 |
| R² pred | 0.1122 | 0.0392 | 0.2810 | 0.1150 | 0.2729 | 0.2297 | 0.7353 | 0.5105 |
| Adequate precision | 8.4989 | 6.7292 | 11.6725 | 9.1737 | 11.7080 | 9.0753 | 24.1321 | 15.8894 |
| Transformation | $\lambda = 1.49$ | no | square root | square root | square root | no | no | no |
| Model type | reduced 2FI | reduced quadratic | reduced 2FI | reduced quadratic | reduced 2FI | reduced quadratic | reduced quadratic | reduced quadratic |

and white increased predicted % budbreak (Fig 1A). For Rich 16−6, the largest effects were White and Red x Far-red; white increased % budbreak and Red and Far-red reduced % budbreak, respectively (Fig 2A).

## Scion shoot growth

The length of scion bud shoots ranged from 60–498 mm for Carrizo and 10–310 mm for Rich 16−6; the wide range indicated that light affected the length of scion bud shoots. The overall models were significant for Carrizo (p < 0.0001) and Rich 16−6 (p = 0.0016), indicating significant factor effects on leaf area. A summary of the ANOVA, lack-of-fit test, three R² statistics, and adequate precision statistic for rootstock diameter is presented (Table 4). A reduced 2FI model was selected for Carrizo and a reduced quadratic model for Rich 16−6. Per the Box-Cox analyses no data transformations were required. The lack-of-fit (LOF) test for Carrizo was significant (p = 0.0004) and was interpreted to be largely a consequence of small error estimates rather than an indication that additional variation could be removed with a better model, as evidenced by the pure error mean square being substantially smaller than both the lack-of-fit and model mean squares (S2 File). The LOF for Rich 16−6 was not significant and indicated that additional variation in the residuals could not be removed with a better model. For Carrizo and Rich 16−6 the three R² statistics were clustered with a difference less than

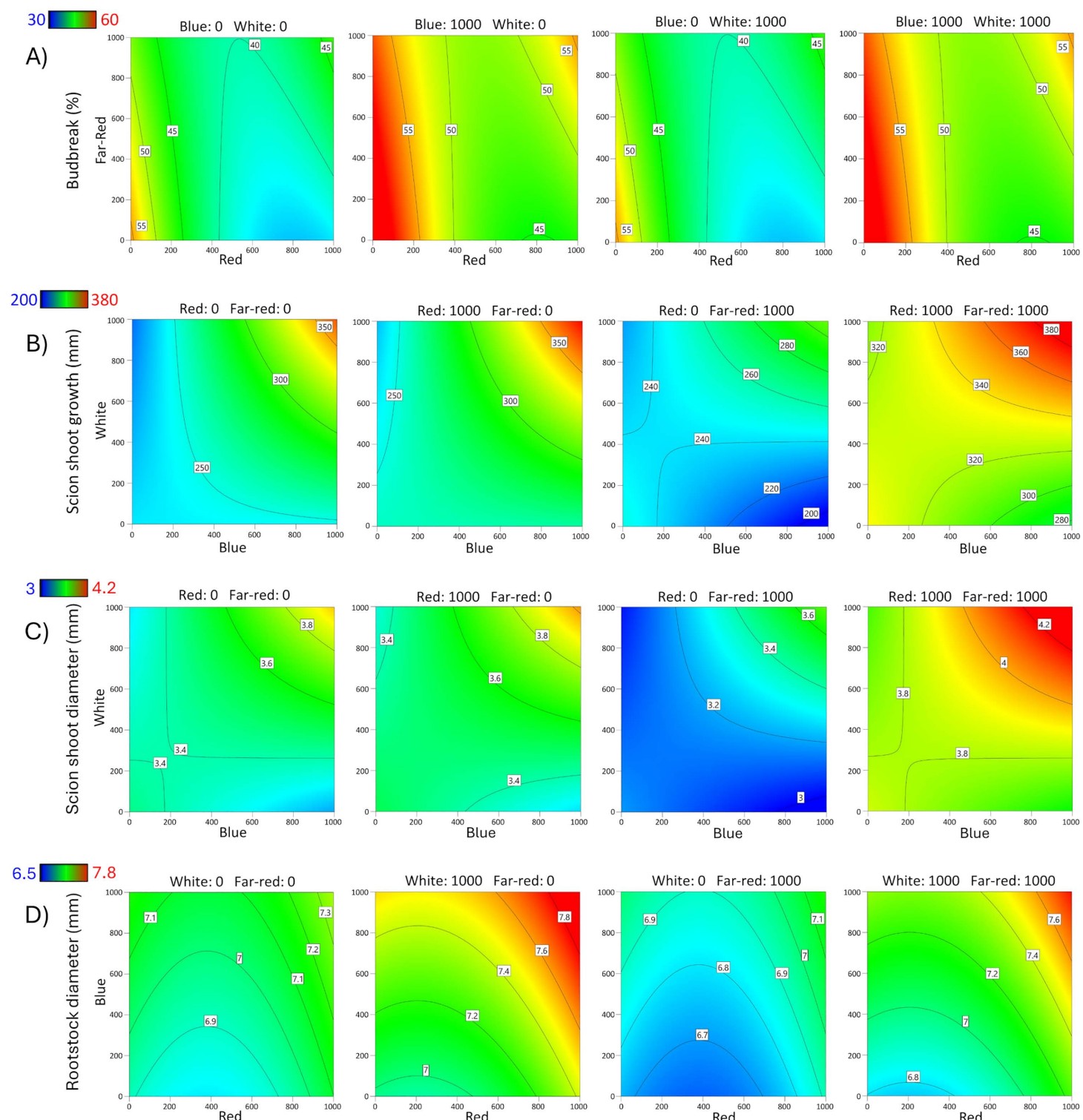

**Fig 1. Contour response surface plots for Carrizo rootstock.** Effects of red, blue, white, and far-red LED light spectra on **A)** Budbreak percentage, **B)** Scion shoot growth, **C)** Scion shoot diameter, and **D)** Rootstock diameter.

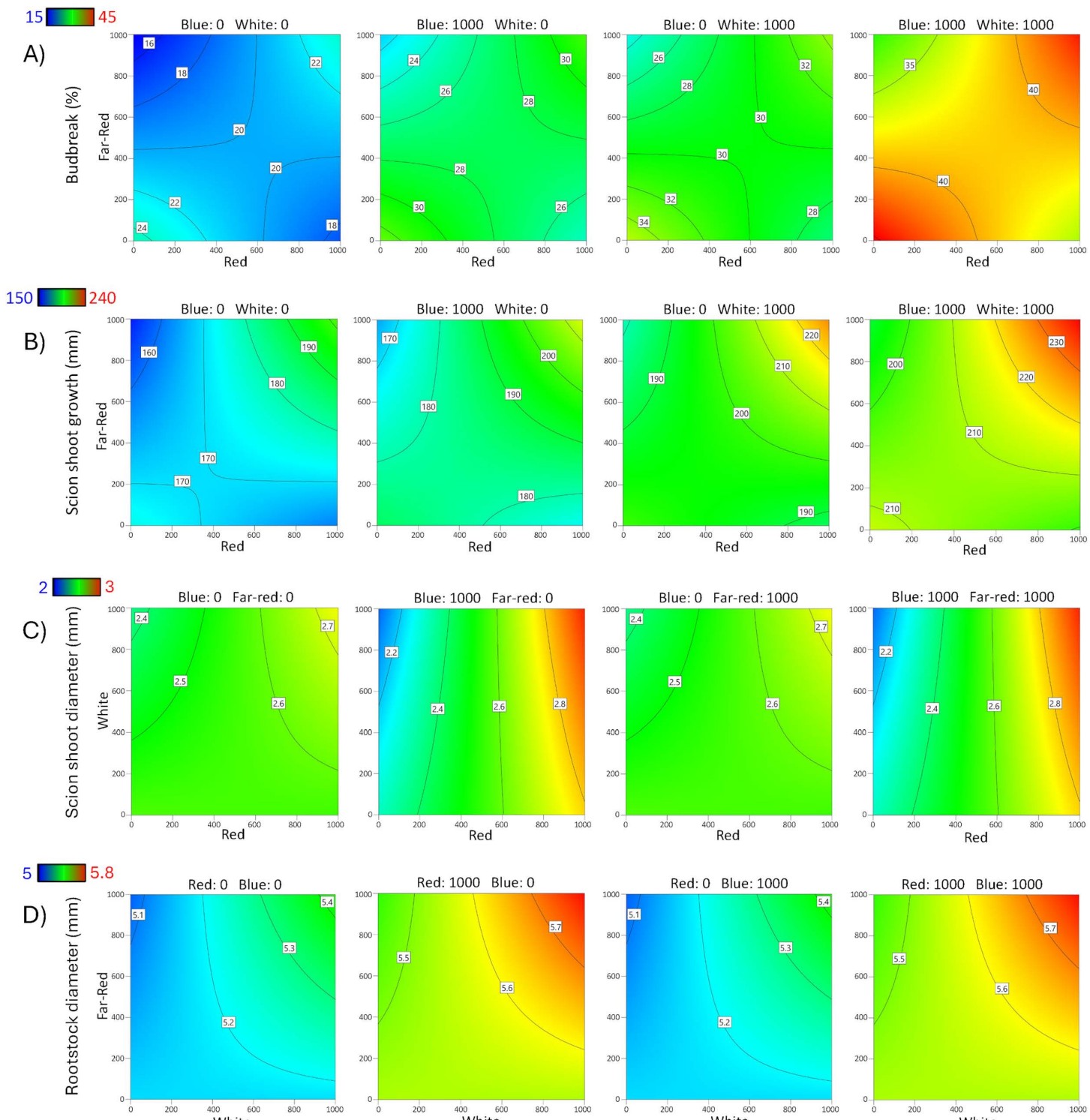

**Fig 2. Contour response surface plots for Rich 16−6 rootstock.** Effects of red, blue, white, and far-red LED light spectra on **A)** Budbreak percentage, **B)** Scion shoot growth, **C)** Scion shoot diameter, and **D)** Rootstock diameter.

0.4 between $R^2$ and $R^2_{predicted}$, and the adequate precision statistics were greater than 4 as recommended for prediction. The ANOVA for Carrizo contained 5 significant terms ($p < 0.05$) – Red, Blue, White, Red x Far-red, and Blue x White. The model predicted that the greatest scion bud growth occurs at high levels Red, Blue, White and Far-red (Fig 1). The ANOVA for Rich 16−6 contained 3 significant terms ($p < 0.05$) – White, Red x Far-red, and Blue$^2$. The model predicted that the greatest scion bud growth occurs at high levels of Red, White, Blue and Far-red (Fig 2).

### Scion shoot diameter

The diameter of shoots growing from the grafted buds ranged from 1.6–5.1 mm for Carrizo and 0.8–4.1 mm for Rich 16−6; the wide range indicated that light affected the diameter of the bud shoots on both rootstocks. The overall models were significant for Carrizo ($p < 0.001$) and Rich 16−6 ($p < 0.001$), indicating significant factor effects on bud shoot diameter. A summary of the ANOVA, lack-of-fit test, three $R^2$ statistics, and adequate precision statistic for bud shoot diameter is presented (Table 4). A reduced 2-factor interaction (2FI) polynomial model was selected for Carrizo and a reduced quadratic model for Rich 16−6. Per the Box-Cox analyses no data transformations were required. The lack-of-fit (LOF) tests were significant for Carrizo ($p = 0.0074$) and not significant for Rich 16−6 ($p = 0.8313$). The significant LOF for Carrizo was interpreted to be largely a consequence of small error estimates rather than an indication that additional variation could be removed with a better model, as evidenced by the pure error mean square being substantially smaller than both the lack-of-fit and model mean squares (S2 File). For Carrizo and Rich 16−6 the three $R^2$ statistics were clustered with a difference less than 0.4 between $R^2$ and $R^2_{predicted}$, and the adequate precision statistics were greater than 4 as recommended for prediction. The ANOVA for Carrizo contained 4 significant terms ($p < 0.05$) – Red, White, Red x Far-red, and Blue x White. The ANOVA for Rich 16−6 contained 2 significant terms ($p < 0.05$) – Red and Blue$^2$. The models for both rootstocks indicate that maximum scion shoot diameter occurs at the maximum LED factor settings (Fig 1, Fig 2) which resulted in 18% and 41% larger diameters for Carrizo and Rich 16−6, respectively, compared to the no supplemental light treatment.

### Rootstock diameter

The diameter of the rootstocks ranged from 6.1–8.7 mm for Carrizo and 3.8–6.1 mm for Rich 16−6; the wide range indicated that light affected the diameter of the rootstocks. The overall models were significant for Carrizo ($p < 0.001$) and Rich 16−6 ($p < 0.001$), indicating significant factor effects on rootstock diameter. A summary of the ANOVA, lack-of-fit test, three $R^2$ statistics, and adequate precision statistic for rootstock diameter is presented (Table 4). A reduced quadratic model was selected for Carrizo and a reduced 2FI model for Rich 16−6. Per the Box-Cox analysis no data transformations were required. The lack-of-fit (LOF) tests were not significant for Carrizo and Rich 16−6 and indicated that additional variation in the residuals could not be removed with a better model. For Carrizo and Rich 16−6 the three $R^2$ statistics were clustered with a difference less than 0.4 between $R^2$ and $R^2_{predicted}$, and the adequate precision statistics were greater than 4 as recommended for prediction. The ANOVA for Carrizo contained 5 significant terms ($p < 0.05$) – Red, Blue, White, Far-red, and Red$^2$. The greatest effects were White ($p < 0.001$) and Red ($p < 0.001$) that predicted an increased diameter of 6%. The ANOVA for Rich 16−6 contained 2 significant terms ($p < 0.05$) – Red and White. The models show that the largest rootstock diameter occurs at low intensity Far-red and high intensity Red, White, and Blue (Fig 1, Fig 2) with an increase in predicted diameter of 12%.

### Internode length

The internode lengths of bud shoots ranged from 5.8–24.2 mm for Carrizo and 3.3–21.5 mm for Rich 16−6; the wide range indicated that light affected internode length. The overall models were significant for Carrizo ($p < 0.001$) and Rich 16−6 ($p = 0.0034$), indicating significant factor effects on internode length. A summary of the ANOVA, lack-of-fit test, three $R^2$ statistics, and adequate precision statistic for rootstock diameter is presented (Table 5). A reduced 2FI model was selected

for Carrizo and a reduced quadratic model for Rich 16−6. Data was transformed for Carrizo per violation of the ANOVA assumption of normally distributed residuals as detected via a Box-Cox analysis. For Rich 16−6 the Box-Cox analysis indicated that no data transformation was required. The lack-of-fit (LOF) test for Carrizo was significant (p = 0.0056) and was interpreted to be largely a consequence of small error estimates rather than an indication that additional variation could be removed with a better model, as evidenced by the pure error mean square being substantially smaller than both the lack-of-fit and model mean squares (S2 File). For Carrizo and Rich 16−6 the three $R^2$ statistics were clustered with a difference less than 0.4 between $R^2$ and $R^2_{predicted}$, and the adequate precision statistics were greater than 4 as recommended for prediction. The ANOVA for Carrizo contained 3 significant terms (p < 0.05) – Red, Blue, and Blue x White. The model shows that the longest internodes were observed at high levels of Blue and White (Fig 3) with a predicted increase in length of 11% compared to no supplemental light. The ANOVA for Rich 16−6 contained 3 significant terms (p < 0.05) – Red, Red x White and $Blue^2$. The model showed a positive association between Red, White, and Blue and internode length (Fig 4).

## Leaf area

Leaf area of bud shoots ranged from 56–1,041 cm$^2$ for Carrizo and 50–478 cm$^2$ for Rich 16−6; the wide range indicated that light affected leaf area. The overall models were significant for Carrizo (p < 0.001) and Rich 16−6 (p < 0.001), indicating significant factor effects on leaf area. A summary of the ANOVA, lack-of-fit test, three $R^2$ statistics, and adequate precision statistic for rootstock diameter is presented (Table 5). A reduced 2FI model was selected for Carrizo and a reduced quadratic model for Rich 16−6. Data were transformed for Carrizo and Rich 16−6 per violation of the ANOVA assumption of normally distributed residuals as detected via Box-Cox analysis. The lack-of-fit (LOF) test for Carrizo was significant (p = 0.0002) and was interpreted to be largely a consequence of small error estimates rather than an indication that additional variation could be removed with a better model, as evidenced by the pure error mean square being substantially smaller than both the lack-of-fit and model mean squares (S2 File). The LOF for Rich 16−6 was not significant and indicated that additional variation in the residuals could not be removed with a better model. For Carrizo and Rich 16−6 the three $R^2$ statistics were clustered with a difference less than 0.4 between $R^2$ and $R^2_{predicted}$, and the adequate precision statistics were greater than 4 as recommended for prediction. The ANOVA for Carrizo contained 5 significant terms (p < 0.05) – Red, Blue, White, Red x Far-red, and Blue x White. The model predicted that the greatest leaf area occurs at high levels Red, Blue and White (Fig 3). The ANOVA for Rich 16−6 contained 4 significant terms (p < 0.05) – Red, White, Red x Far-red, and $Blue^2$. The model predicted that the greatest leaf area occurs at high levels of Red, White, Blue and Far-red (Fig 4).

## Scion dry weight

Scion dry weight ranged from 1–12 g for Carrizo and 0.83–5.3 g for Rich 16−6 (Table of data – supplement); the wide range indicated that light affected scion dry weight. The overall models were significant for Carrizo (p < 0.001) and Rich 16−6 (p < 0.001), indicating significant factor effects on scion dry weight. A summary of the ANOVA, lack-of-fit test, three $R^2$ statistics, and adequate precision statistic for rootstock diameter is presented (Table 5). A reduced 2FI model was selected for Carrizo and a reduced quadratic model for Rich 16−6. Data were transformed for Carrizo per violation of the ANOVA assumption of normally distributed residuals as detected via Box-Cox analysis. The lack-of-fit (LOF) test for Carrizo was significant and was interpreted to be caused by small error estimates rather than an indication that additional variation could be removed with a better model. The LOF for Rich 16−6 was not significant and indicated that additional variation in the residuals could not be removed with a better model. For Carrizo and Rich 16−6 the three $R^2$ statistics were clustered with a difference less than 0.4 between $R^2$ and $R^2_{predicted}$, and the adequate precision statistics were greater than 4 as recommended for prediction. The ANOVA for Carrizo contained 5 significant terms (p < 0.05) – Red, Blue, White, Red x Far-red, and Blue x White. The model shows that the greatest dry weight occurs at high levels of Red, Blue, White, and

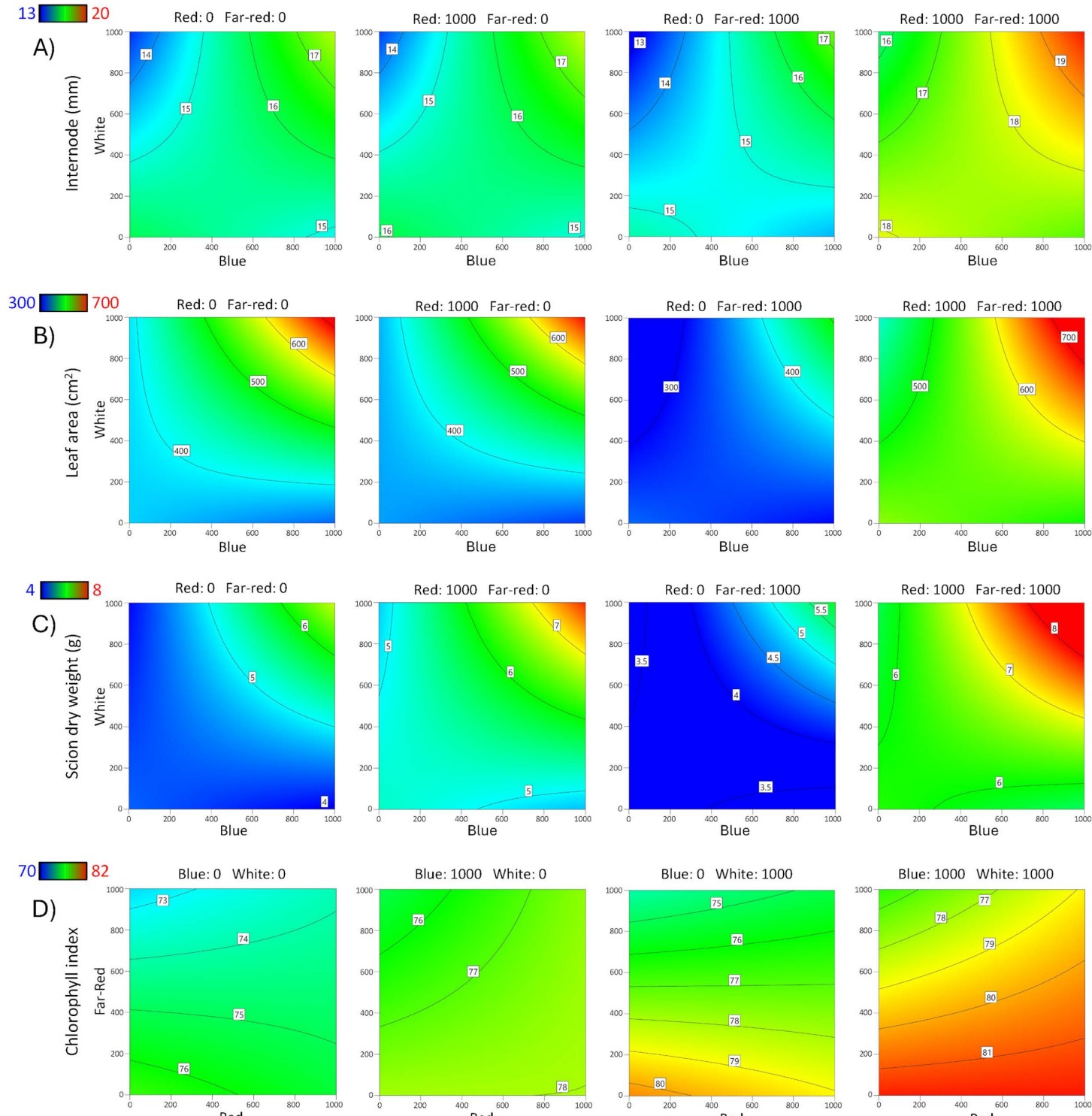

**Fig 3. Contour response surface plots for Carrizo rootstock.** Effects of red, blue, white, and far-red LED light spectra on **A)** Internode length, **B)** Leaf area, **C)** Scion dry weight, and **D)** Chlorophyll index.

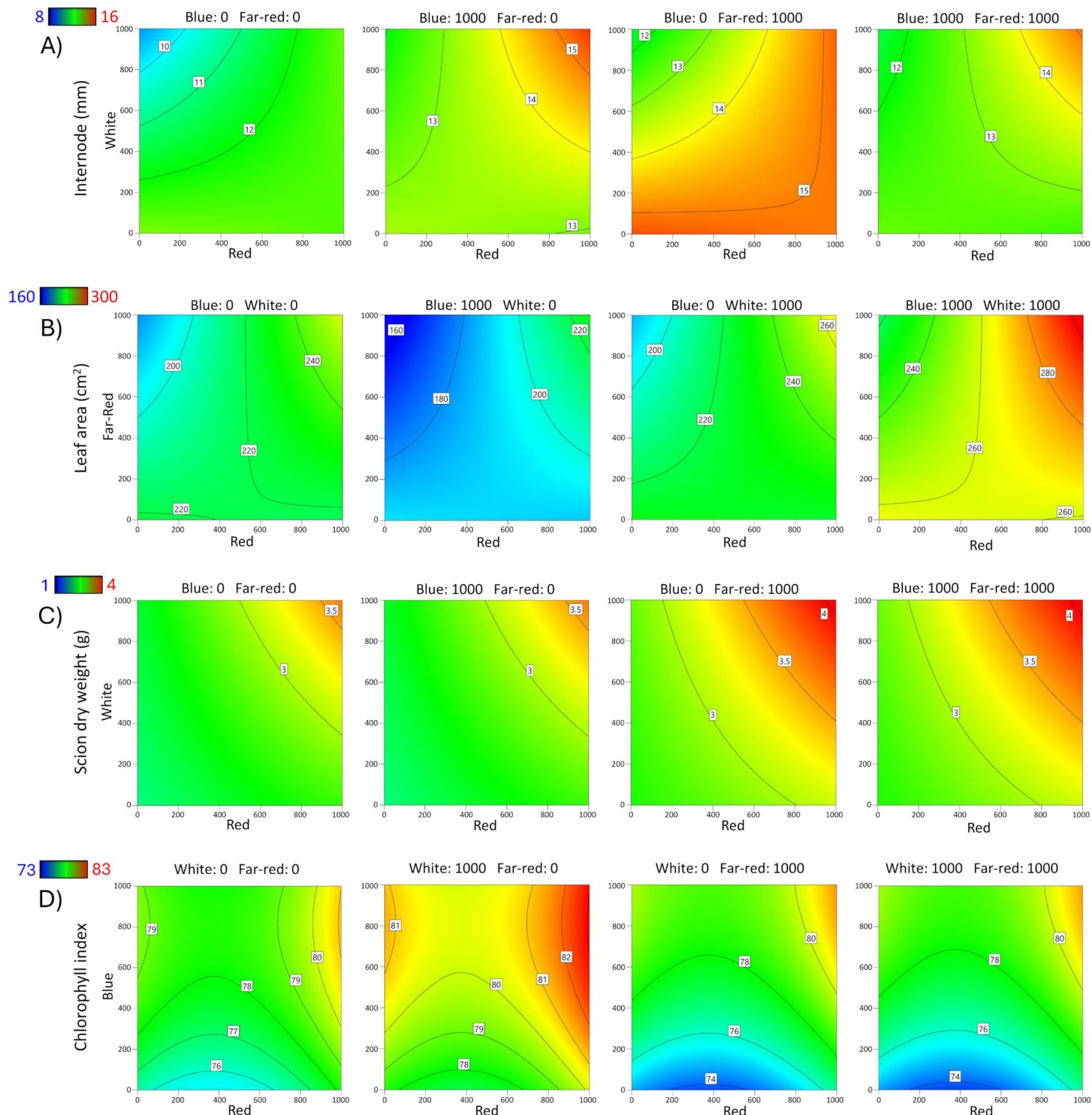

**Fig 4. Contour response surface plots for Rich 16−6 rootstock.** Effects of red, blue, white, and far-red LED light spectra on **A)** Internode length, **B)** Leaf area, **C)** Scion dry weight, and **D)** Chlorophyll index.

Far-red (Fig 3). The ANOVA for Rich 16−6 contained 5 significant terms (p < 0.05) – Red, White, Far-red, Red x White, and Blue$^2$. The model predicted that the greatest dry weight occurs at high levels of Red, White, and Blue (Fig 4).

## Chlorophyll index

The leaf chlorophyll index ranged from 71.6–81.6 for Carrizo and 71.5–82.8 for Rich 16−6 (Table of data – supplement); the wide range indicated that light affected the leaf chlorophyll index. The overall models were significant for Carrizo (p < 0.001) and Rich 16−6 (p < 0.001), indicating significant factor effects on the leaf chlorophyll index. A summary of the ANOVA, lack-of-fit test, three $R^2$ statistics, and adequate precision statistic for rootstock diameter is presented (Table 5). A reduced quadratic model was selected for Carrizo and Rich 16−6. For Carrizo and Rich 16−6 the Box-Cox analyses indicated that no data transformations were required. The lack-of-fit (LOF) tests for Carrizo and Rich 16−6 were significant and were interpreted to be largely a consequence of small error estimates rather than an indication that additional variation could be removed with a better model, as evidenced by the pure error mean square being substantially smaller than both the lack-of-fit and model mean squares (S2 File). For Carrizo and Rich 16−6 the three $R^2$ statistics were clustered with a difference less than 0.4 between $R^2$ and $R^2_{predicted}$, and the adequate precision statistics were greater than 4 as recommended for prediction. The ANOVA for Carrizo contained 8 significant terms (p < 0.05) – Red, Blue, White, Far-red, Red x Blue, Red x Far-red, White x Far-red, and White$^2$. The model predicted that the greatest leaf chlorophyll index measure occurs at high levels of Blue and low levels of Far-red (Fig 3). The ANOVA for Rich 16−6 contained 8 significant terms (p < 0.05) – Red, Blue, White, Far-red, Blue x Far-red, White x Far-red, Red$^2$, and Blue$^2$. The model predicted that the greatest leaf chlorophyll index measures occur at high levels of Red, Blue, and White, and low levels of Far-red (Fig 4).

## Discussion

This study evaluated the effects of supplemental LED light spectra and intensity on the winter growth of grafted citrus nursery plants, focusing on vegetative budbreak and scion shoot development. The findings demonstrate that both the spectral quality and intensity of supplemental LED lighting significantly influence key physiological and morphological parameters in young citrus trees, with implications for optimizing nursery production during periods of limited natural light.

Although the RSM models explained only a modest proportion of the total variation in most growth traits, the three $R^2$ statistics were tightly clustered and adequate precision values were well above the recommended threshold of 4, indicating strong signal-to-noise within the experimental design space. This pattern reflects the biological reality that much of the variability among the grafted citrus trees was due to the inherent plant to plant and micro environmental differences that cannot be removed by adjusting light spectra alone. In practice, the fitted response surfaces are therefore best viewed as tools for identifying directional trends and practical optima in the LED spectra and intensity, rather than for predicting the exact response of individual trees. The models describe how changes in red, blue, white, and far-red intensities shift the expected mean of the various responses across treatments, while recognizing that two trees exposed to the same spectral combination will still differ because of underlying biological noise that is not under control.

### Effects of light spectra on budbreak and growth

The results indicate that blue and white light, particularly at higher intensities, promote vegetative budbreak of the scion on both Carrizo and Rich 16−6 rootstocks, while red and far-red light tend to suppress budbreak when applied at higher intensities. These findings are consistent with previous studies in citrus and other species, where blue light has been shown to stimulate budbreak and enhance the differentiation of axillary buds. For example, higher budbreak percentages were observed in citrus under LED lighting with a pronounced blue peak [3], and similar positive effects of blue light have been reported in wheat [30], plum [31], and rose [32,33]. The present study further supports the role of blue-enriched spectra in breaking dormancy and initiating shoot growth in grafted citrus. The positive effect of white light on budbreak observed here can be attributed to its high proportion of blue wavelengths, as the white LEDs used had a correlated color

temperature (CCT) of 5700K, which is known to emit a broad spectrum with a substantial blue component. This aligns with the understanding that commercial white LEDs often utilize blue diodes with phosphor coatings [34], resulting in a spectrum that includes biologically active wavelengths beyond the blue region [35].

In contrast, red light alone reduced budbreak, a result that mirrors findings in other woody species where red light can suppress the formation of axillary buds and maintain apical dominance. The suppressive effect of red and far-red light on budbreak, particularly at higher intensities, highlights the importance of spectral balance in managing developmental responses in citrus propagation.

## Synergistic effects of combined spectra on shoot growth

While blue and white light were most effective for budbreak, the greatest scion shoot growth was achieved when red, blue, and white light were combined at maximum intensity. The addition of far-red light to this combination further increased shoot elongation and leaf area, although far-red light alone did not enhance growth. This synergistic effect is consistent with the Emerson enhancement effect, where the simultaneous excitation of both photosystem I (PSI) and photosystem II (PSII) by a combination of PAR (400–700 nm) and far-red photons (700–800 nm) increases the quantum yield of photosynthesis. Previous research has shown that far-red light, while inefficient on its own, can enhance photochemical efficiency and leaf-level photosynthetic rates when combined with shorter wavelengths [36–38]. Thus, the Emerson effect's spectral synergy suggests commercial nurseries could add far-red LEDs over existing red/blue arrays to accelerate the shoot elongation. The study also found that far-red light increased leaf area only when combined with red light, and not with other wavelengths, suggesting that specific spectral interactions are necessary to maximize canopy expansion and light interception. This observation aligns with reports in other crops where far-red supplementation promotes leaf expansion and biomass accumulation under controlled environments [13,15,16, 39–46].

## Chlorophyll content and photoprotection

Chlorophyll content was highest under blue and white light and lowest under far-red light, with the chlorophyll index increasing with blue light intensity. These results are in agreement with prior studies showing that blue light is essential for chlorophyll synthesis and prevents etiolation, while red light alone is insufficient [47–51]. The ability of plants to adjust chlorophyll content and the chlorophyll a/b ratio in response to light quality reflects adaptive mechanisms for optimizing photosynthetic efficiency and photoprotection under varying spectral conditions [33,49,50,52–55].

## Rootstock-specific responses and light spectrum correlations

The study identified both shared and rootstock-specific morphological responses to light spectra in Carrizo and Rich 16−6. While blue and white light enhanced vegetative budbreak in both rootstocks, inhibitory effects from red and far-red light were more pronounced in Carrizo. After bud emergence, all tested spectra positively influenced subsequent bud development, though spectral associations with specific growth traits differed. Once buds exceeded 2 mm, continued growth occurred under all spectral combinations, but trait optimization required wavelength-specific combinations: red light enhanced scion dry weight despite suppressing initial budbreak, far-red light reduced chlorophyll content but synergized with other spectra to increase leaf area, and white light had the strongest correlation with rootstock trunk growth, particularly in Carrizo, where it had the single largest effect. Being broad-spectrum, white light likely optimized photoassimilate allocation. In contrast, far-red light (735 nm) showed the highest correlation with internode elongation, especially in Rich 16−6, where its interaction with red light increased internode length by 18% compared to far-red-deficient spectra, an effect consistent with far-red's known role in phytochrome-mediated shade avoidance responses where the red-to-far-red (R:FR) light ratio regulates shoot elongation [56]. These results indicate the need for phase-specific spectral management—prioritizing blue/white light for dormancy release, then specific red-far-red ratios for canopy or rootstock

development. Rootstock genotype further modulated responses, with Carrizo showing greater sensitivity to far-red during the early growth stage.

## Implications and future directions

The results of this study have practical implications for commercial citrus nurseries. By optimizing supplemental LED light spectra and intensity, it is possible to enhance budbreak and shoot growth during winter, potentially reducing the duration of a production cycle and improving propagation efficiency. The findings support a multivariate approach to light management, considering both the main effects and interactions among different wavelengths.

Future research should explore the underlying physiological mechanisms driving these responses, including hormonal regulation and gene expression associated with light perception and signaling. Additionally, field validation of these findings under commercial nursery conditions would help refine best practices for supplemental lighting. Further investigation into cultivar-specific and rootstock-specific responses to light spectra could enable more targeted management strategies. Natural daylight DLI increased over the course of the winter experiment and contributed to the total DLI. All treatments received the same amount of natural sunlight. Treatment differences were solely from the imposed supplemental LED spectra and intensities; PPFD measurements were taken at night to avoid interference from daylight and obtain accurate, treatment-specific PPFD measures for each LED spectrum. Thus, varying natural daylight did not confound comparisons among the LED treatments in this study, although the absolute magnitudes and potentially some quantitative aspects of the response surfaces could differ under substantially higher or lower background DLI than the 5–11 mol m$^{-2}$ day$^{-1}$ range observed during this winter trial.

## Conclusion

This study demonstrated that supplemental LED lighting, with controlled spectra and intensity, can enhance the winter growth of grafted citrus nursery plants. By extending the photoperiod to 16 hours and manipulating the proportions of blue, red, white, and far-red wavelengths, it is possible to influence both vegetative budbreak and scion shoot development in young citrus trees. Blue and white light, particularly at higher intensities, were most effective in promoting budbreak, while the greatest shoot elongation and leaf area were achieved when all four spectra were combined at maximum intensity.

The responses observed were generally consistent across the two rootstocks tested, Carrizo and Rich 16−6, suggesting that these lighting strategies may be broadly applicable within citrus nursery production. These findings support the use of multivariate approaches to optimize supplemental lighting, considering both the main effects and interactions among different wavelengths.

The results provide a practical basis for improving nursery efficiency during periods of limited natural light, potentially reducing production time and enhancing propagation success. However, the study was conducted under specific controlled greenhouse conditions, and further research is warranted to validate these findings under a range of commercial nursery environments and across a wider range of citrus genotypes.

Overall, this work contributes to the understanding of how precise manipulation of light quality and intensity can be used to improve citrus nursery production and provides guidance for the development of best practices in supplemental lighting management.

## Supporting information

**S1 Fig. Spectral power distribution (SPD) profiles.** Runs 1–11.
(TIF)

**S2 Fig. Spectral power distribution (SPD) profiles.** Runs 12–23.
(TIF)

**S3 Fig. Spectral power distribution (SPD) profiles.** Runs 24–29.
(TIF)

**S1 File. Light treatments and phenotypic data for Carrizo and Rich 16−6 rootstocks.** Excel workbook with three worksheets summarizing the light treatment design and raw phenotypic measurements for Carrizo and Rich 16−6 across all experimental treatments.
(XLSX)

**S2 File. ANOVA Tables and Response Surface Regression Models.** ANOVA tables and model equations in both coded and actual forms. Coded equations use factor ranges from −1 to +1, allowing easy comparison of effect sizes, while actual equations predict responses at specific factor settings.
(DOCX)

## Acknowledgments

We would like to thank Sailindra Patel and Kerry Worton for their help with tree manipulations, data collection, and sample collection and analysis.

## Author contributions

**Conceptualization:** Randall P. Niedz, Rayane Barcelos Bisi, Ute Albrecht, Kim D. Bowman.

**Formal analysis:** Randall P. Niedz, Rayane Barcelos Bisi.

**Investigation:** Rayane Barcelos Bisi.

**Methodology:** Randall P. Niedz, Rayane Barcelos Bisi, Ute Albrecht, Kim D. Bowman.

**Project administration:** Kim D. Bowman.

**Writing – original draft:** Randall P. Niedz, Rayane Barcelos Bisi.

**Writing – review & editing:** Randall P. Niedz, Rayane Barcelos Bisi, Ute Albrecht, Kim D. Bowman.

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
