## [Decision Letter · Decision Letter 0]

15 Dec 2025

PONE-D-25-45421Effects of LED light spectra and intensity on winter citrus nursery productionPLOS One

Dear Dr. Niedz,

Thank you for submitting your manuscript to PLOS ONE. After careful consideration, we feel that it has merit but does not fully meet PLOS ONE’s publication criteria as it currently stands. Therefore, we invite you to submit a revised version of the manuscript that addresses the points raised during the review process.

We look forward to receiving your revised manuscript.

Kind regards,

Ali Akbar Ghasemi-Soloklui, Ph.D

Academic Editor

PLOS One

**Journal Requirements:**

1. When submitting your revision, we need you to address these additional requirements. Please ensure that your manuscript meets PLOS ONE's style requirements, including those for file naming. The PLOS ONE style templates can be found at https://journals.plos.org/plosone/s/file?id=wjVg/PLOSOne_formatting_sample_main_body.pdf and https://journals.plos.org/plosone/s/file?id=ba62/PLOSOne_formatting_sample_title_authors_affiliations.pdf 2. Thank you for stating the following financial disclosure: This research was conducted at the United States Horticultural Research Laboratory, Ft Pierce, Florida, USA and was funded by the California Citrus Nursery Board (ALB-21-23 and BOW-21-23).   Please state what role the funders took in the study.  If the funders had no role, please state: "The funders had no role in study design, data collection and analysis, decision to publish, or preparation of the manuscript." If this statement is not correct you must amend it as needed. Please include this amended Role of Funder statement in your cover letter; we will change the online submission form on your behalf. 3. Please upload a new copy of Figures 1 to 4 and S1 to S3, as the detail is not clear. Please follow the link for more information:  https://journals.plos.org/plosone/s/figures 4. If the reviewer comments include a recommendation to cite specific previously published works, please review and evaluate these publications to determine whether they are relevant and should be cited. There is no requirement to cite these works unless the editor has indicated otherwise.

Reviewers' comments:

Reviewer's Responses to Questions

**Comments to the Author**

1. Is the manuscript technically sound, and do the data support the conclusions?

Reviewer #1: Yes

Reviewer #2: Partly

Reviewer #3: Partly

Reviewer #4: Partly

Reviewer #5: Yes

2. Has the statistical analysis been performed appropriately and rigorously? 

Reviewer #1: Yes

Reviewer #2: No

Reviewer #3: Yes

Reviewer #4: N/A

Reviewer #5: Yes

3. Have the authors made all data underlying the findings in their manuscript fully available?

Reviewer #1: Yes

Reviewer #2: Yes

Reviewer #3: Yes

Reviewer #4: Yes

Reviewer #5: Yes

4. Is the manuscript presented in an intelligible fashion and written in standard English?

Reviewer #1: Yes

Reviewer #2: Yes

Reviewer #3: Yes

Reviewer #4: No

Reviewer #5: Yes

5. Review Comments to the Author

**Reviewer #1:**  This manuscript provides a comprehensive study on the impact of supplemental LED lighting on growth of grafted citrus nursery plants during winter. The research is thorough and offers valuable insights, there are some areas that needs to be improved.

Strength:

1. The use of four factor D optimal response surface design is robust approach to analyze interaction between light spectra and intensity. This multivariate method provides detailed insights into the effects of different light combinations.

2. The study measures eight plant growth and physiological responses, providing a holistic view of how light affects citrus nursery production.

Scope for Improvements:

1. The study focuses on two rootstocks, Carrizo and Rich 16-6, which seemed to be differentially responded to light spectra in a few of parameters tested. (Table 3&4, orange highlights). Presenting Two-way ANOVA for interaction between light spectra and rootstock can strengthen study outcomes for general recommendations.

2. Results for scion shoot growth are missing in the text.

3.Some models showed significant lack-of-fit (e.g., Carrizo budbreak and scion shoot diameter), which may indicate that the models do not fully capture the variability in the data. While the authors attribute this to small error estimates, further refinement of the models could improve reliability.

Additional comments are provided the PDF.

**Reviewer #2:** The manuscript requires substantial restructuring of the methodology, improved clarity, stronger statistical justification, significantly enhanced figures & tables formatting and a more rigorous discussion grounded in the actual strength of the presented data.

**Reviewer #3:** 1. End the introduction with a clear statement of why this study matters.

2. missing details on sampling design, system boundaries, assumptions, and analytical procedures.

3. Results lack consistency in units, numerical values, and figure/table referencing, reducing scientific reliability.

4. The conclusion is broad and not aligned with the stated goals, failing to synthesize the most important findings or implications.

5. No figures from your conducted experiment?

6. The study does not quantify uncertainty (sensitivity analysis, error ranges, variability in input data), which weakens the reliability of the results and violates ISO-LCA transparency requirements.

**Reviewer #4:**  The manuscript presents a solid investigation into LED light spectra and intensity effects on grafted citrus nursery growth using response surface methodology (RSM), addressing a practical gap in winter propagation efficiency. Strengths include the large sample size (2,784 plants), comprehensive measurements (8 responses), and multivariate design capturing light interactions, with clear biological insights like blue/white light promoting budbreak and full-spectrum enhancing shoot growth. The paper has merit for PLOS ONE due to its applied value and data availability, but requires major revisions: reanalyze with covariate-adjusted models (e.g., including DLI), report effect sizes/CIs, verify assumptions rigorously, and add power calculations.

Some questions:

1. Does the study adequately address potential confounding effects of varying natural daylight DLI (ranging from 8-11 mol m⁻² day⁻¹ across treatments) on the observed LED supplemental light responses, given that PPFD measurements were taken at night?

2. Why were atypical seedlings removed based solely on visual morphology rather than genetic or polyembryony confirmation, and how might this selection bias affect rootstock uniformity?

3. Is the inverted T-budding method and south-facing bud orientation standardized sufficiently across 2,784 plants to rule out micro-environmental variations in budbreak initiation?

4. Given multiple significant lack-of-fit (LOF) tests (e.g., p=0.0166 for Carrizo budbreak, p=0.0004 for scion growth), why were these dismissed as due to "small error estimates" without exploring higher-order terms or alternative models?

5. For models with low R² values (e.g., budbreak R²=0.1733 for Carrizo), does the adequate precision >4 justify predictive claims, and were effect sizes or confidence intervals reported to contextualize practical significance?

6. Why was forward selection via AICc used for ANOVA model building instead of exhaustive or stepwise methods, and were multicollinearities among light intensities (Red, Blue, White, Far-red) assessed via VIF before including interactions?

7. With data transformations applied selectively (e.g., square root for leaf area), were post-transformation normality and homoscedasticity formally verified for all responses using Shapiro-Wilk and Breusch-Pagan tests?

8. For the D-optimal design with only 5 replicates for pure error, is the power sufficient to detect biologically meaningful effect sizes, and were multiple comparison adjustments applied across the 8 responses?

**Reviewer #5:**  well done. may you make infographic for the experimental design will enhance this work. in the future research, may you try different types of light such as, red, blue, purple..................

...etc.

6. PLOS authors have the option to publish the peer review history of their article (what does this mean? ). If published, this will include your full peer review and any attached files.

**Do you want your identity to be public for this peer review?** For information about this choice, including consent withdrawal, please see our Privacy Policy .

Reviewer #1: No

Reviewer #2: **Yes:** Ashraf Muhammad Arslan

Reviewer #3: No

Reviewer #4: No

Reviewer #5: **Yes:** Mahmoud Adel Ahmed Ali

---

## [Author Response · Author response to Decision Letter 1]

9 Feb 2026

Editor

3. Please upload a new copy of Figures 1 to 4 and S1 to S3, as the detail is not clear. Please follow the link for more information: https://journals.plos.org/plosone/s/figures

Figures 1-4 and S1-S3 are 300 dpi Tiff files formatted per PLoS figure guidelines. The figures are blurry as displayed in the rendered pdf but when downloaded via the link in the rendered file they are clear.

Reviewer #1

1. The study focuses on two rootstocks, Carrizo and Rich 16-6, which seemed to be differentially responded to light spectra in a few of parameters tested. (Table 3&4, orange highlights). Presenting Two-way ANOVA for interaction between light spectra and rootstock can strengthen study outcomes for general recommendations.

Thank you for highlighting the potential value of examining the interaction between light spectra and rootstock. Analyzing as a two-way ANOVA is not possible given the way the study was constructed. Light spectra were not varied as a categorical treatment with four fixed levels. Instead, each light spectrum was varied over a quantitative range within a response surface methodology framework rather than as discrete categorical levels as for a two-way ANOVA.

In the spirit of your comment, it is possible to include rootstock as a factor (discrete, 2 levels) in the RSM analysis. The result would be a 5-factor RSM with 4 quantitative factors (the 4 light spectra) and 1 categoric factor (the 2 rootstocks). As you indicate, analyzing in this manner would provide rootstock × (continuous light variables) interaction terms to determine whether Carrizo and Rich 16-6 differ in their response surfaces across the studied light conditions.

The issue with analyzing this way is that rootstock had a very large effect and including it as a factor would dominate the explained variance and drive the overall R2 well above 0.90. High R2 values would look impressive but could be a bit misleading since they would mainly reflect the between-rootstock shift rather than a good description of the within rootstock response surface to light conditions. For example, the R2 was 0.17 for Budbreak % for Carrizo and 0.20 for Rich 16-6, but when rootstock is added as a factor the R2 of the model jumps to 0.97 and the p-value for the main effect of rootstock is 8.9 x 10-186!

In this study, the primary question was the light–response relationship within each rootstock (i.e., the response surface over the quantitative spectral factors), not simply the overall difference between Carrizo and Rich 16 6. Because rootstock is a very strong factor it would largely summarize the obvious mean shift between rootstocks while obscuring the more subtle, but biologically important, effects and interactions among the continuous light variables. We think that by analyzing each rootstock separately preserved the interpretability of the light–response surfaces and avoided over-emphasizing an inflated overall R^2 that would mostly represent the strong rootstock main effect rather than the patterns that are most relevant for optimizing light management.

More details of why the 2 rootstocks were analyzed separately were added to the beginning of the Results section.

2. Results for scion shoot growth are missing in the text.

Missing scion shoot growth section added – Added - thanks for catching this!

3.Some models showed significant lack-of-fit (e.g., Carrizo budbreak and scion shoot diameter), which may indicate that the models do not fully capture the variability in the data. While the authors attribute this to small error estimates, further refinement of the models could improve reliability.

We acknowledge that the models may not fully capture the variability in the data. Higher order cubic models do improve the fit to the data, but the experimental design was specifically constructed to estimate parameters up to quadratic terms only. Fitting a higher-order model beyond the design's capacity introduces the risk of overfitting.

Given that the pure error MS was substantially lower than both the lack-of-fit MS and model MS in all of the analyses explains some of the significant LOF. The extremely significant LOF for Chlorophyll index is almost certainly due to an extremely small pure error MS vs LOF and model MS. First, we added the full ANOVA tables to supplement File S2 for each measured response. This allows the reader to see all the ANOVA values. Second, we included the observation that the pure error MS was substantially lower than both the lack-of-fit MS and model MS in all the analyses and is consistent as a possible explanation for the significant LOF.

Additional comments are provided the PDF. (points addressed in the PDF or directed to the response to reviewers)

Reviewer #3

1. End the introduction with a clear statement of why this study matters.

Rewrote the last paragraph to more clearly explain the purpose of the study and why it matters.

2. missing details on sampling design, system boundaries, assumptions, and analytical procedures.

3. Results lack consistency in units, numerical values, and figure/table referencing, reducing scientific reliability.

4. The conclusion is broad and not aligned with the stated goals, failing to synthesize the most important findings or implications.

5. No figures from your conducted experiment?

6. The study does not quantify uncertainty (sensitivity analysis, error ranges, variability in input data), which weakens the reliability of the results and violates ISO-LCA transparency requirements.

We thank the reviewer for these observations and fully agree that the issues raised (e.g., clear methods, consistency of results reporting, alignment of conclusions with goals, inclusion of figures, and transparency about uncertainty) are critical for scientific rigor. However, because the comments are expressed in very general terms and do not point to specific sections or examples, it is challenging to respond in a precise way. We would gratefully welcome more detailed guidance so that we can revise the manuscript accordingly.

Reviewer #4

Some questions:

1. Does the study adequately address potential confounding effects of varying natural daylight DLI (ranging from 8-11 mol m⁻² day⁻¹ across treatments) on the observed LED supplemental light responses, given that PPFD measurements were taken at night?

This paragraph was added at the end of the Discussion addressing this point –

Natural daylight DLI increased over the course of the winter experiment and made up part of the total DLI, with all treatments affected equally – received the same amount of natural sunlight. Treatment differences were solely from the imposed supplemental LED spectra and intensities; PPFD measurements were taken at night to avoid interference from daylight and obtain accurate, treatment-specific PPFD measures for each LED spectrum. Thus, varying natural daylight did not confound comparisons among the LED treatments in this study, although the absolute magnitudes and potentially some quantitative aspects of the response surfaces could differ under substantially higher or lower background DLI than the 5–11 mol m⁻² day⁻¹ range observed during this winter trial.

Further, the LED light treatments section in Materials and Methods was revised to more clearly explain how the supplemental lighting was applied. Supplemental LEDs were used only to make up the difference between the natural daylength and a total of 16 hours. For example, if the natural daylength was 10 hours, the LED lights were turned on 6 hours before sunrise and then switched off at sunrise.

2. Why were atypical seedlings removed based solely on visual morphology rather than genetic or polyembryony confirmation, and how might this selection bias affect rootstock uniformity?

Citrus rootstock production relies on nucellar polyembryony as the source of rootstock seed. Off-type seedlings originating from zygotic embryos exhibit distinct morphological characteristics that differ markedly from nucellar seedlings - altered leaf morphology, growth habit, and inferior vigor. Commercial citrus nurseries routinely remove seedlings exhibiting clear morphological deviations—typically those less than 30% the height of typical seedlings or displaying aberrant leaf characteristics. Studies examining commercial nursery populations confirm that visual rogueing effectively eliminates zygotic seedlings, with residual very low off-type frequencies in field plantings of most rootstocks.

A sentence (and a referenced study) was added to the Materials section explaining that citrus rootstock seed is produced by nucellar polyembryony and rogueing off types is the standard practice in commercial citriculture.

3. Is the inverted T-budding method and south-facing bud orientation standardized sufficiently across 2,784 plants to rule out micro-environmental variations in budbreak initiation?

These sentences were added to the Materials section to explain the control of micro-environment effects –

The inverted T-budding method and south-facing bud orientation were implemented with a high degree of standardization, supported by multiple procedural and design controls. The uniform budding date, pre-budding plant selection, randomization, continuous environmental monitoring, and high replication collectively minimize the influence of micro-environmental variation on budbreak initiation.

4. Given multiple significant lack-of-fit (LOF) tests (e.g., p=0.0166 for Carrizo budbreak, p=0.0004 for scion growth), why were these dismissed as due to "small error estimates" without exploring higher-order terms or alternative models?

See response to Reviewer #1, Point 3 on this point.

5. For models with low R² values (e.g., budbreak R²=0.1733 for Carrizo), does the adequate precision >4 justify predictive claims, and were effect sizes or confidence intervals reported to contextualize practical significance?

This is a good point that is not adequately addressed, but should be, in the manuscript. A complete explanation of how to interpret a model with low to modest R2 but adequate precision was added to the start of the Discussion section. Adding this section should be quite helpful. Although the RSM models explained only a modest proportion of the total variation in most growth traits, the three R² statistics were tightly clustered and adequate precision values were well above the recommended threshold of 4, indicating strong signal to noise within the experimental design space. This pattern reflects the biological reality that much of the variability among the grafted citrus trees was due to the inherent plant to plant and micro environmental differences that cannot be removed by adjusting light spectra alone. In practice, the fitted response surfaces are therefore best viewed as tools for identifying directional trends and practical optima in the LED spectra and intensity, rather for predicting the exact response of individual trees. The models describe how changes in red, blue, white, and far-red intensities shift the expected mean of the various responses across treatments, while recognizing that two trees exposed to the same spectral combination will still differ because of underlying biological noise that is not under control.

6. Why was forward selection via AICc used for ANOVA model building instead of exhaustive or stepwise methods, and were multicollinearities among light intensities (Red, Blue, White, Far-red) assessed via VIF before including interactions?

We appreciate this thoughtful question and the opportunity to clarify our approach.

We chose forward selection with AICc because it identifies RSM models where the goal is accurate prediction and surface estimation rather than formal hypothesis testing of every possible term. An issue with looking at multiple model reduction methods and selecting the “best” one, in our view, amounts to a form of p hacking (unless all of the models are presented), so we committed a priori to using forward selection via AICc given that it is a method well-established for RSM. We did not begin with a main effects only model and then add interactions in a separate step.

VIFs were assessed via several measures including Leverage, Cook’s distance, and DFFITS plots. Also, there were some indirect protections and measures of multicollinearities. The way D-optimal RSM designs are constructed minimizes multicollinearity among the factors. All models exceeded the adequate precision threshold of 4, indicating sufficient signal strength despite any potential collinearity. R2 clustering – indicated model stability and an absence of severe multicollinearity, which would typically cause a large divergence between these R2 measures. Added Leverage and DFFITS to the description in the M&M of model influence diagnostics used.

7. With data transformations applied selectively (e.g., square root for leaf area), were post-transformation normality and homoscedasticity formally verified for all responses using Shapiro-Wilk and Breusch-Pagan tests?

While formal tests such as Shapiro–Wilk (for normality) and Breusch–Pagan (for homoscedasticity) are one way to assess ANOVA assumptions, we used a residual diagnostic workflow that is a standard approach in RSM analysis that examines the same assumptions as the Shapiro–Wilk and Breusch–Pagan tests, but does so through graphical diagnostics and transformation rather than separate omnibus tests on each response. The model diagnostic procedure we used included a Box–Cox analysis to determine whether a transformation was required, inspection of normal probability plots of residuals, plots of internally studentized residuals versus predicted values to verify, respectively, approximate normality and homoscedasticity after transformation. We also examined lack of fit tests, Cook’s distance plots, leverage plots, DFFITS plots, the agreement among R², adjusted R², and predicted R², and required adequate precision values > 4. These diagnostics, taken together, indicated that the transformed models met the ANOVA assumptions sufficiently for valid inferences.

Rewrote the description of the model diagnostics and adequacy tests rationale and workflow.

8. For the D-optimal design with only 5 replicates for pure error, is the power sufficient to detect biologically meaningful effect sizes, and were multiple comparison adjustments applied across the 8 responses?

Added to the Experimental design and analyses section in the M&M

In planning the study, we conducted a prospective power analysis of the selected 4 factor D optimal RSM design assuming a residual standard deviation of 1 on the coded response scale. Under this assumption, the design had very high power (≥ 99.6% at α = 0.05) to detect the prespecified main, interaction, and quadratic effect sizes: for example, the computed powers for A, B, C, D and their two factor interactions (AB, AC, AD, BC, BD, CD) were all 99.9%, and the powers for the quadratic terms A², B², C², and D² were 99.9%, 99.9%, 99.9%, and 99.6%, respectively. These effect sizes correspond to changes that are large relative to the within treatment variability and are biologically meaningful for nursery performance, indicating that the design is well powered for effects of practical interest.

We followed a response surface analysis strategy for how assumptions and multiple outcomes are handled that differs from the univariate, multiple testing–oriented approach. The eight responses are considered different but related aspects of plant performance, therefore a single multiple comparison correction is not used; instead, each model is examined separately with a focus on effects that were clearly estimated and that pointed in the same direction across related measures.

Added to the Experimental design and analyses section in the M&M

The RSM framework used in this study differs from the traditional univariate approach used where each outcome is analyzed in isolation: a separate set of formal tests run for each response, and the results interpreted one outcome at a time. In contrast, the RSM approach still fits a separate model to each response but places greater emphasis on overall model adequacy—assessed via residual plots, lack of fit, and summary statistics such as R², adjusted R², predicted R²,

---

## [Decision Letter · Decision Letter 1]

7 Apr 2026

Effects of LED light spectra and intensity on winter citrus nursery production

PONE-D-25-45421R1

Dear Dr. Niedz,

We’re pleased to inform you that your manuscript has been judged scientifically suitable for publication and will be formally accepted for publication once it meets all outstanding technical requirements.

Kind regards,

Eugenio Llorens

Academic Editor

PLOS One

Additional Editor Comments (optional):

Reviewers' comments:

Reviewer's Responses to Questions

**Comments to the Author**

1. If the authors have adequately addressed your comments raised in a previous round of review and you feel that this manuscript is now acceptable for publication, you may indicate that here to bypass the “Comments to the Author” section, enter your conflict of interest statement in the “Confidential to Editor” section, and submit your "Accept" recommendation.

Reviewer #2: All comments have been addressed

Reviewer #5: All comments have been addressed

2. Is the manuscript technically sound, and do the data support the conclusions?

Reviewer #2: Yes

Reviewer #5: Yes

3. Has the statistical analysis been performed appropriately and rigorously? 

Reviewer #2: Yes

Reviewer #5: Yes

4. Have the authors made all data underlying the findings in their manuscript fully available?

Reviewer #2: Yes

Reviewer #5: Yes

5. Is the manuscript presented in an intelligible fashion and written in standard English?

Reviewer #2: Yes

Reviewer #5: Yes

6. Review Comments to the Author

Reviewer #2: (No Response)

Reviewer #5: for this research well done. thanks for taking comment in consideration .

7. PLOS authors have the option to publish the peer review history of their article (what does this mean? ). If published, this will include your full peer review and any attached files.

**Do you want your identity to be public for this peer review?** For information about this choice, including consent withdrawal, please see our Privacy Policy .

Reviewer #2: **Yes:** Ashraf Muhammad Arslan

Reviewer #5: **Yes:** Mahmoud Ali

---

## [Editor Report · Acceptance letter]

PONE-D-25-45421R1

PLOS One

Dear Dr. Niedz,

I'm pleased to inform you that your manuscript has been deemed suitable for publication in PLOS One. Congratulations! Your manuscript is now being handed over to our production team.

Kind regards,

on behalf of

Dr. Eugenio Llorens

Academic Editor

PLOS One